# Enzymes Involved in Antioxidant and Detoxification Processes Present Changes in the Expression Levels of Their Coding Genes under the Stress Caused by the Presence of Antimony in Tomato

**DOI:** 10.3390/plants13050609

**Published:** 2024-02-23

**Authors:** Francisco Luis Espinosa-Vellarino, Inmaculada Garrido, Ilda Casimiro, Ana Cláudia Silva, Francisco Espinosa, Alfonso Ortega

**Affiliations:** 1Grupo Investigación Fisiología y Biología Celular y Molecular de Plantas (BBB015), Facultad de Ciencias, Campus Avenida de Elvas s/n, Universidad de Extremadura, 06071 Badajoz, Spain; flespinosav@unex.es (F.L.E.-V.); igarridoc@unex.es (I.G.); casimiro@unex.es (I.C.); aortegagarrido@unex.es (A.O.); 2Centro Tecnológico Nacional Agroalimentario “Extremadura” (CTAEX), Ctra. Villafranco-Balboa 1.2, 06195 Badajoz, Spain; asilva@ctaex.es

**Keywords:** antimony, ascorbate, glutathione, glutathione reductase, glutathione-S-transferase, tomato

## Abstract

Currently, there is an increasing presence of heavy metals and metalloids in soils and water due to anthropogenic activities. However, the biggest problem caused by this increase is the difficulty in recycling these elements and their high permanence in soils. There are plants with great capacity to assimilate these elements or make them less accessible to other organisms. We analyzed the behavior of *Solanum lycopersicum* L., a crop with great agronomic interest, under the stress caused by antimony (Sb). We evaluated the antioxidant response throughout different exposure times to the metalloid. Our results showed that the enzymes involved in the AsA-GSH cycle show changes in their expression level under the stress caused by Sb but could not find a relationship between the *NITROSOGLUTATHIONE REDUCTASE* (*GSNOR*) expression data and nitric oxide (NO) content in tomato roots exposed to Sb. We hypothesize that a better understanding of how these enzymes work could be key to develop more tolerant varieties to this kind of abiotic stress and could explain a greater or lesser phytoremediation capacity. Moreover, we deepened our knowledge about Glutathione S-transferase (GST) and Glutathione Reductase (GR) due to their involvement in the elimination of the xenobiotic component.

## 1. Introduction

Plants are sessile organisms that can live in adverse conditions, where their growth and development are severely affected [1]. Due to this, plants have had to evolve and develop defense mechanisms that allowed them to survive and adapt to the changing environment [2,3]. Many of them, including crop species, are exposed to different stresses (biotic and abiotic or the combination of the two) in open fields [4]. Climate change is accelerating the appearance of symptoms caused by these severe conditions [5].

Heavy metals (HM) and metalloids are elements with an atomic mass and specific gravity higher than 20 and 5, respectively, and they are found in nature as subcomponents of the earth’s crust [6,7]. The contamination caused by these elements is a global problem for human health and the environment and, therefore, can cause abiotic stress to plants. Currently, the danger is greater due to their increased presence in fertile soils at high levels. These high levels can be mainly explained by anthropogenic activities, among others, becoming a potential risk for agricultural production, as they affect crop growth, development, and yield [7].

HM can be classified into essential and non-essential according to their biological importance and potential effect on plant growth and performance. Some examples of essential HM arecopper (Cu), nickel (Ni), manganese (Mn), iron (Fe), and zinc (Zn), which are required by plants at low concentrations as micronutrients and have a role in plant metabolism pathways at optimal concentrations [6,8,9]. The intensive bioavailability of these essential nutrients primarily triggers an oxidative burst that alters plant physiology and biochemistry [10]. Mainly, HM or metalloid toxicity negatively affects root and shoot development and seed germination in plants [11,12]. On the other hand, these types of elements affect cell division and elongation, nutrient absorption and metabolism, transpiration rate, photosynthetic pigment synthesis, crop productivity, and fruit quality [13,14] and inactivate several key enzymes and proteins to hinder essential metal ion substitution reactions of biomolecules [7,15]. All these alterations can lead to the death of the plant [14]. On the other hand, under stress conditions, plant cells respond by generating Reactive Oxygen Species (ROS), such as superoxide anion (O_2_^−^), hydrogen peroxide (H_2_O_2_) and the hydroxyl ion (^·^OH). These compounds can be generated in different cellular compartments such as chloroplasts, mitochondria, peroxisomes, endoplasmic reticulum and plasmalemma or the apoplast [14]. ROS play a role as second messengers or signaling molecules in cell development and in holding cell and organism homeostasis [16]. However, the accumulation of ROS can be toxic, since it generates an imbalance in redox homeostasis and can produce oxidative stress that damages biomolecules [17,18]. In addition to ROS, when the plant is under stress, there is also an alteration of reactive nitrogen species (RNS): nitric oxide (NO), peroxynitrite (ONOO-) and S-nitrosoglutathione (GSNO) [19]. An increase in the amount of RNS at the cellular level causes nitrosative stress which, combined with ROS, leads to nitro-oxidative stress [20,21,22]. Consequently, plants have developed enzymatic and non-enzymatic antioxidant systems to efficiently remove ROS [23]. Several studies have reported a strong increase in ROS and RNS production in plants due to HM and/or metalloid toxicity [24,25,26,27]. Due to this, plants have developed detoxification mechanisms to overcome metal toxicity by modulating signaling molecules, antioxidant enzyme activities, redox status, phytochelatin content, and the flavonoid biosynthetic pathway [7,28].

The metalloid antimony (Sb) is a non-essential trace element for plants which does not prevent it from being absorbed in its soluble forms [29]. However, absorption capacity depends on the plant species and the bioavailability of Sb [30]. Depending on the redox state of the soil, Sb can be found in the form of antimonite (Sb [III]) (more toxic) or antimonate (less toxic) (Sb [V]) [31]. Once Sb is absorbed by the plant, it can accumulate in the root or in the leaf tissue [32,33]. Sb is highly toxic to animals and plants and, in the former, its toxicity is due to its ability to bind to thiol functional groups on proteins [34]. In turn, the reactions of Sb with the thiol groups of glutathione or phytochelatins is a mechanism used by plants as a detoxification mechanism [35]. Particularly, Sb toxicity causes a redox imbalance that alters the enzymatic activity of superoxide dismutase (SOD), peroxidase (POX), ascorbate peroxidase (APX), dehydroascorbate reductase (DHAR), glutathione reductase (GR), Glutathione S-transferase (GST) and nitrosoglutathione reductase (GSNOR) [27,36,37]. It has also been reported that Sb toxicity stress involves the collaboration of phenolic compounds, flavonoids, phenylpropanoid glycosides (PPG), carotenoids, and components of the ascorbate–glutathione (AsA/GSH) cycle, all these compounds constituting the non-enzymatic antioxidant pathway [27,35,38,39]. The increase in these ROS and RNS could interact with the activity of antioxidant mechanisms, leading to the modification of gene expression in genes involved in plant defense systems [20,22].

Plants can act as natural bioaccumulators since they are capable of absorbing and accumulating different HM and metalloids from soil and water [9]. However, not all plants have the same response to HM/metalloid toxicity. There are some species, such as *Dittrichia viscosa* L., that tolerate or accumulate high concentrations of Sb without suffering oxidative damage [40]. This species can grow in soils contaminated by HM and/or metalloids, acting as a phytoremediator [40].

Tomato (*Solanum lycopersicum* L.) is an annual crop that belongs to the Solanaceae family, and it is used as a model plant for its agronomic interest and its widespread cultivation throughout the world. So far, only a few studies have focused on demonstrating its tolerance/capacity to develop in contaminated soils by HM with results on the changes occurring at the enzymatic and molecular level during the detoxification processes [41,42].

The aim of this work is to deepen the knowledge of how the defensive reactions that happen in tomato plants evolve over time when they are under Sb stress. Our experimental design was aimed at determining if there are expression changes in several genes involved in the detoxification processes as the exposure time to the metalloid increased. There are data on the expression of enzymes involved in defense and detoxification processes at specific times of exposure to Sb, but there are not so many reports on how the expression pattern of genes encoding enzymes involved in defense activities and detoxification evolves. We carry out an evaluation of the expression level of the genes involved in the antioxidant activity of *APX*, *DHAR*, *GR*_*cyt*, *GR*_*chl* (participating in the AsA-GSH cycle) and *GST*, *GST TCHQD*, *SOD* and *GSNOR* under stress by metalloid antimony in different concentrations (0, 0.5 mM and 1.0 mM) throughout time at different intervals (1, 7 and 14 days after exposure to Sb, das). In addition, young leaves are analyzed to determine whether the expression pattern differs between plants without stress and plants exposed to Sb and to compare these results with the differences found in other research where adult leaves are the target.

## 2. Results

### 2.1. Phenotypic Evaluation

We first aimed to observe the phenotypic appearance of the plants exposed to Sb at different concentrations (0.0, 0.5 and 1.0 mM Sb) and at the final exposure time (14 days after exposure). The plants exposed to the metalloid at different concentrations showed chlorosis and necrosis in the leaves, while the control plants did not show this phenotype (Figure 1). Furthermore, plants exposed to 1.0 mM Sb presented severe damage (Figure 1C). When analyzing the roots, the Sb-treated plants showed less development and changes in the color (the roots turned brownish) in comparison with the control roots (compare Figure 1A to Figure 1B,C). The Sb content in the plant at t3 (in shoots and in the root) as well as the translocation factor showed a similar result to that obtained in a previous experiment (Appendix A).

In addition, we wanted to investigate whether the exposure to the metalloid led to differences in the leaf area. We confirmed that the plant aerial part area decreased as the concentration of Sb increased (Figure 2).

### 2.2. Analysis of the Expression Levels of Genes Encoding Enzymes Involved in Antioxidant Defense

As the plants exposed to Sb showed different phenotypes compared to the non-exposed ones, we decided to investigate whether detoxification processes were occurring and, therefore, whether genes encoding enzymes involved in antioxidant defense were changing their expression when the plants were under stress [43,44]. For that reason, we assessed the Sb effect in the expression levels of genes encoding key enzymes involved in those processes at different time points (1, 7 and 14 days after exposure; t1, t2 and t3, respectively) and using different Sb concentrations (0.0, 0.5 and 1.0 mM Sb) in tomato leaves and roots. The results are described for each group of enzymes below.

### 2.3. AsA/GSH Enzymes: APX, DHAR and GR

At t2 and t3, we observed that there was a higher expression of *APX* in the root than in the leaf compared to its controls (Figure 3). At t2 and t3, the 0.5 mM treatment showed a significant difference, with Sb-free plants showing higher and lower expression in leaves and roots, respectively, compared to the treated ones.

As observed in Figure 3, *DHAR* reaches its highest expression rate at the final time of exposure (t3; 14 days after exposure) in the root. The same expression pattern was not observed on the leaf, except for the leaves from plants treated with 0.5 mM Sb at t3. Interestingly, at t1, the plant leaves treated with 0.5 mM Sb showed a significant reduction in the expression level when compared to the control, while at t3, the opposite behavior was observed, with a higher expression of Sb-treated plants.

For both *GR* coding genes isoforms, *GR_cyt* and *GR_chl*, at t3, the roots treated with Sb showed, in general, higher expression than the non-treated roots (Figure 3), but only significant differences were encountered in the 0.5 mM Sb treatment for the *GR_cyt* coding gene (Figure 3). In leaves, at the latest measurement point, we observed the opposite pattern, with significance in the 0.5 mM Sb treatment for *GR_chl* (Figure 3). It was also interesting to see a reduction in the expression levels in both organs using the two different Sb concentrations (0.5 and 1.0 mM) for the *GR_chl* isoform at t2 (Figure 3).

### 2.4. SOD and GST

We observed that there was a higher expression of *SOD* at t2 and t3 in the root than in the leaf when compared to their controls, as we analyzed in *APX* (Figure 3 and Figure 4). Like the expression pattern of *APX*, at t3, the 0.5 mM treatment showed a significant difference, with Sb-free plants showing a lower and higher expression in roots and leaves, respectively, when compared to the treated ones. Moreover, at t3, *SOD* revealed the same expression pattern for plants treated with 1.0 mM Sb (Figure 4).

For *GST* isoforms, significant differences were found in leaves and roots at 1 day after exposure (t1). In leaves, the genes coding these enzymes are more expressed than in the control ones (Figure 4), except for 1.0 mM Sb-treated leaves in the *GST TCHQD* isoform. Meanwhile, in roots, a lower expression in the treated leaves is encountered (Figure 4). At the final exposure time (t3), we observed that the *GST TCHQD* isoform presented lower expression levels in the 0.5 and 1.0 mM-treated leaves and 1.0 mM-treated roots when compared to the non-exposed controls. Also, we realized that roots treated with 1.0 mM Sb presented a basal expression level in all time points for the same *GST* isoform (Figure 4).

### 2.5. Interaction of GSNOR Expression and Root NO Content

Considering the results obtained for the *GSNOR* expression levels, we observed a significant reduction in those levels for leaves treated with 0.5 mM Sb when compared to non-treated leaves at medium and late exposures (Figure 5).

Although no significant changes were found in the expression levels in roots, we could observe NO content increase in roots exposed to the metalloid at 7 and 14 das, using medium and high Sb concentrations (Figure 6A,B). Hence, it seems that in the studied system no relation can be found between *GSNOR* expression and NO content in roots under Sb-induced stress.

### 2.6. Protein Interaction Analysis

We analyzed the interactions between the enzymes under study using the STRING v11.0 tool. According to the results (Figure 7), we report that there is a close relationship between DHAR, GST, GR_cyt, GR_chl, APX, and SOD. GR and APX interact with the largest number of enzymes, while GSNOR (with GR_chl and APX) and GST TCHQD (only with DHAR) are the enzymes that interact the least with the rest of the enzymes analyzed. On the one hand, it seems that there is a high degree of interaction between GST, GR_chl and DHAR and that GST acts as an intermediary. Moreover, APX could have a similar role to GST in the GR_chl-APX-SOD relationship. Furthermore, APX interacts with GST, DHAR and GR_cyt, GSNOR, but to a lesser extent than with GR_chl and SOD.

### 2.7. Phylogenetic Analysis and GST and GR Isoform Peptide Sequence and Motif Analysis

At the phylogenetic level, a close relationship between Solanaceae GR and *Arabidopsis thaliana* L. GR is clearly observed (Figure 8A). Specifically, GR1_*A.thaliana* is phylogenetically close to GR_cyt and GR_*A.thaliana* to Solanaceae GR_Chl (Figure 8A). Having analyzed GST isoforms in the Solanaceae family, we observed a relationship between Solanaceae GST and *Arabidopsis* GST TAU 19, while Solanaceae GST TCHQD had a relation with *A. thaliana* GST Phi 8 (Figure 8B).

GR performance depends on the presence of a number of specific domains: the FAD-binding domain (stabilizes the sulfhydryl (-SH) group), the NADPH-binding domain (where NADPH-dependent reduction in GSSG to GSH occurs), the GSSG-binding complex and an interface domain connecting the different subunits in dimeric GR [45,46]. Having analyzed the alignment of peptide sequences (Figure 9), we found a region of the FAD binding site in all GRs. This domain is conserved in all species, except in the pepper chloroplast GR, where we observed some divergences. A high degree of conservation was also determined in both the active site and the GSH-binding domain, except for *Arabidopsis* GR2. Although we found differences in the cytosolic domains, no orthologs were found in *Capsicum annuum* L. GR chloroplast and in both *Solanum tuberosum* L. (potato) GRs.

There are different classes of GST described in *A. thaliana* such as GSTU τ (Tau), GSTF Φ (Phi), GSTL (Lambda), GSTT (Theta), GSTZ (Zeta), DHAR, TCHQD (tetrachlorohydroquinone dehalogenase), EF1Bγ (elongation 1B, hemerythrin and Iota) [47]. The GST Tau and GST Phi isoforms are the ones that have great implication in the detoxification processes of xenobiotic compounds and are the most abundant in plants [48]. The Tau isoform is the most relevant in plant resistance to oxidative stress factors due to its GSH and GPOX-dependent dextification activity [49]. GST Tau has a series of residues described in both *A. thaliana* and *Oryza sativa* L. spp. *japonica*, dicot and monocot model species, respectively. GST Tau keeps the residues described in *Arabidopsis* (dicot) and *Oryza sativa* (monot) such as residues involved in domain interactions or residues involved in GSH binding. These residues are also found in the GST isoforms analyzed from tomato and other Solanaceae (Figure 10). Among the conserved residues, we highlight two arginine residues (Arg, R): Arg20, responsible for the protonation of GSH and Arg98, which intervenes in the establishment of a hydrogen bond that offers stability to the molecule. These residues in *Oryza sativa* (OsGST_TAU_23) are very important because their removal leads to the loss of its catalytic region [50].

In our research, we analyzed two GST isoforms described in Solanaceae: GST and GST TCHQD. Both isoforms collaborate in detoxification processes of xenobiotic compounds, and there is evidence of their relevant role in the defense of plants against oxidative stress induced by HM or metalloids [41,51]. Regarding the TCHQD isoform, it has been reported that in *Sphingobium chlorophenolicum,* it catalyzes the reductive dehalogenation of TCHQ and trichlorohydroquinone to degrade pentachlorophenol [52].

All the residues involved in functional or binding interactions with GSH of the GST TAU are conserved in *Arabidopsis*, *Oryza sativa* and *Solanaceae* (Figure 10). However, this information is lacking in *Zea mays* L., for which ortholog information is absent.

## 3. Discussion

### 3.1. Phenotypic Evaluation

Sb toxicity induces stunted growth and development of the aerial part and roots in different plant species [27,51,53,54,55]. However, in other species such as *Z. mays* or *Helianthus annuus* L., slight alterations or even no effect on growth have been described [24,56], and in *Brassica napus* L. and *Brassica rapa* L. concentrations of 8 mg/kg Sb in the soil have a positive effect on growth [57]. Such slight or even beneficial Sb-induced alterations have been described at much lower concentrations than those used in this work.

In the present study, tomato plants showed symptoms consistent with those described in tomato and *D. viscosa* under Sb stress in previous experiments [40,58]. At the foliar level, chlorosis and necrosis were observed in plants exposed to the metalloid compared to control plants, which was more noticeable in plants exposed to 1.0 mM Sb. Similar results have been reported in plant leaves under As-induced stress [59]. Plant roots exposed to Sb showed brownish tones when compared to control roots, which has also been reported by other authors using Sb or other metalloids in different plants [40,58,59]. In plants under Sb toxicity, lesser root development and a clear decrease in the biomass at the aerial part were observed when compared to the control plants. Leaf area decreased significantly with increasing Sb concentration in the growing medium. These results are similar to those described by Peško et al. [60] and Espinosa-Vellarino et al. [41] under Sb stress, and to those described by Bharti and Sharma [59], in which growth retardation, inhibition of root extension, reduction in leaf number and leaf senescence, and consequently biomass decline were also observed, which could lead to plant death.

### 3.2. Analysis of the Expression Levels of Genes Encoding Enzymes Involved in Antioxidant Defense

APX is an enzyme of the AsA-GSH cycle, but together with SOD, it is a key component of the antioxidant defense system in plant cells [61], and these components may work sequentially. On the one hand, SOD dismutates the superoxide radical (O_2_^−^) to hydrogen peroxide (H_2_O_2_) via the Haber–Weiss reaction [62]. Different SOD isoforms are distributed in different cell components (chloroplast, mitochondrion, nucleus, peroxisome, cytoplasm and apoplast). SOD is classified into three groups according to the metal cofactor present in the active site. Thus, Mn-SOD is located in mitochondria and peroxisomes, Cu/Zn-SOD can be found in the cytosol, mitochondria, plastids and in the apoplast, and Fe-SOD is frequently located in chloroplasts, cytosol, mitochondria and peroxisomes [63,64,65]. Furthermore, APX belongs to the superfamily of haemoperoxidases [43] that can be located at the cytosolic, chloroplastidial (thylakoidal, stromal) and peroxisomal levels [66]. Its function is to reduce H_2_O_2_ to H_2_O (using ascorbate as an electron acceptor), and it also plays an important role in plant growth [44,67,68], in the regulation of seed vigor [69] and in the senescence process in rice [70]. Based on our results, we observed that antioxidant enzymes APX and SOD after t1 show a similar pattern. In fact, at t2 and t3, a higher expression of both enzymes in the root than in the leaf was observed. Moreover, the expression levels in the root were higher than those detected in the control plants, but not in the leaves. Expression levels in this organ were always higher in control plants, which can be explained by the fact these enzymes are involved in vegetative growth in control plants, while in plants exposed to Sb their expression is restricted to the root zone to mitigate the damage caused by this metalloid. However, at t1, the two enzymes showed different expression patterns, where *APX* is already more expressed in the root and even in the leaves in plants exposed to the metalloid, although only significant differences were found in the expression levels in leaves exposed to 1.0 mM Sb. These expression results are consistent with those obtained for enzyme activity in Espinosa-Vellarino et al. [41] at 14 days (t3) of Sb-induced stress in the root. SOD activity at t3 was higher at higher Sb concentrations, which as not noted for APX, where there was more expression at 0.5 mM Sb than at 1.0 mM. The differences in expression that we observed in leaves in both studies may be due to the great variability observed in both cases, although the greater expression in roots than in leaves was also noticed in other studies where Sb [27,36,71] or other HM [25,72,73] were also used. *APX* and *SOD* expressions are highly controlled by different abiotic stresses such as drought, salinity, extreme temperatures or the presence of metals [44,67,68,74]. Therefore, both activities should have a similar range of expression. *Arabidopsis* is known to improve its tolerance to metal or metalloid (Zn, Cd, Al, Cu and As) toxicity stress when *SOD* and *APX* genes are overexpressed [75,76]. Consequently, these enzymes are important in the plant response to this stress. Although in tomato, the obtained response shows a different expression of *APX* and *SOD* between leaves and roots, which could also be observed in *Dittrichia*, where there was a higher activity of these antioxidant enzymes in both parts of the plant. In both plant species, Sb tends to accumulate in the root, which leads to a higher expression of *APX* and *SOD* in roots when compared to leaves [27,73]. This restriction of Sb in the root may avoid damage to flowers and leaves that could compromise reproductive and photosynthetic events in plants, respectively [77,78]. This has also been reported in previous research with Zn and Pb, where traces of these compounds were found in the roots, but not in the ovules and the embryo sac, so that the seeds were free of toxic compounds [77]. This retention in the root zone could be due to the Caspary band that acts by disrupting the apoplast at the level of the endodermis [77,78].

DHAR is an enzyme of the AsA-GSH cycle, and it is crucial in AsA recycling catalyzing the GSH-dependent reduction in dehydroascorbate (DHA) [79]. DHAR isoforms act in organelles such as chloroplasts, mitochondria, peroxisomes and cytosol [79]. Their activity is essential for regulating both apoplastic and symplastic AsA reserves, maintaining cellular redox homeostasis. *DHAR* expression increases as the time of exposure to Sb increases, at least in the root (significant at t3). In the leaf, we did not observe the same pattern of expression, except for 0.5 mM at t3. The treatment with 1.0 mM Sb maintained the level of expression throughout the time studied. As with APX, there is a lot of variability, especially with 1.0 mM Sb. Under oxidative stress conditions, several studies indicate that there is an increase in DHAR activity that leads to a better tolerance to stress conditions such as salinity and drought [80,81]. In fact, overexpression of *DHAR* genes in transgenic plants resulted in increased levels of AsA in tissues, conferring enhanced tolerance to stresses induced by cold, water deficit, salinity and HM toxicity [17,79]. Moreover, significant differences in total AsA content were observed in *DHAR1*, *DHAR2* and *DHAR3* knockout mutants in *Arabidopsis*, confirming the need for increased DHAR activity to reduce DHA during stress [82]. In addition, Bashir and John [83] described how applying Si together with brassinosteroid (BR; 1 mM) improved the tolerance of tomato seedlings under cold stress. This response could be due to increased expression of genes encoding enzymes involved in antioxidant defense, such as *CAT*, *MDHAR*, *DHAR* and *GR* [83]. Therefore, our results highlight the importance of DHAR against abiotic stress. However, other studies show a considerable decrease in DHAR activity under HM or metalloid stress. This is the case in As-treated *Oryza sativa* and *Triticum* family, where APX and MDHAR activities increased while DHAR decreased [84,85]. In contrast, in *Dittrichia*, an increase in DHAR activity has been found in both leaf and root in Sb-stressed plants [58]. We can state that, although the expression of *DHAR* in tomato under Sb-induced stress has a similar pattern to that of the activity of this enzyme described for *Dittrichia*, a plant with good remediation ability, its behavior is different if we compare it with the activity of this enzyme in tomato, rice or wheat, plants that are not considered good remediators [41,84,85]. In *H. annuus*, Sb treatment also significantly increased DHAR activity, especially in roots, which possibly contributed to antioxidant defense [27]. Considering the expression results and the previous results obtained for enzymatic activity, we can deduce that, although there is a higher *DHAR* expression to increase the reduction in DHA to AsA using GSH [86], its activity is conditioned by several factors. On the one hand, DHAR is susceptible to high concentrations of H_2_O_2_, changing its activity. Considering the high levels of *SOD* and *APX* expression, we suppose that upon Sb toxicity, the amount of this ROS is very high and may interfere with the functional capacity of DHAR. In addition, the binding of Sb to GSH may occur as a detoxifying mechanism, and therefore, when the concentration of Sb increases, there is a higher GSH consumption, even if more is being synthesized which can be explained by increased *GR* expression and activity. The increase in *DHAR* expression does not result in increased activity, possibly due to the lack of available GSH once it is forming GSH-Sb complexes.

To avoid the toxicity of HM or metalloids such as Sb, plants synthesize different chelating compounds such as organic acids, glutathione (GSH), phytochelatin (PC), class I and II metallothionein (MT), nicotianamine, but also heat shock proteins (HSP), proline or even phytohormones [87,88].

Glutathione (GSH) takes part in the non-enzymatic antioxidant mechanism, and it works as a substrate for enzymes such as DHAR. It is a tripeptide with a thiol group that allows it to capture metals by acting as a detoxifying agent. GSH can reduce Cu^2+^ uptake in rice as it promotes the action of antioxidant enzymes and the glyoxalase system [89]. In addition, GSH is a cellular antioxidant and a signal molecule for ROS, which can be scavenged by GSH, and collaborates in the maintenance of the cellular redox state [90,91]. The enzymes responsible for GSH synthesis (GR, glutathione peroxidase (GPX)) or those that collaborate for its conjugation with xenobiotic elements (GST) actively participate in the defense of plants against oxidative stress [51,88,92]. Glutathione can be found in two different states: reduced (GSH) or oxidized (glutathione disulfide (GSSG)) and it is the reduced state that acts as an antioxidant as, in this state, the thiol group of cysteine can either donate an electron that reduces ROS by neutralizing them or interact with the HM or metalloid by chelating it [87,88]. This electron transfer from GSH to a wide range of xenobiotic compounds is catalyzed by GST, allowing a non-toxic conjugate to be generated and stored in vacuoles. In fact, several studies show that GSH and metabolizing enzymes such as GST protect plants against oxidative stress [51,92]. When GSH loses an electron, it is more feasible that it can interact with another GSH forming oxidized compound GSSG. In order to recover its reducing capacity, GSSH must be transformed into GSH by the GR enzyme, another enzyme of the AsA-GSH cycle that has been analyzed and employing the reducing power of NADPH. GSH has great importance in oxidative stress conditions. Under these conditions, GR is continuously activated to regenerate GSH [93]. It has been suggested that the GSH/GSSG ratio is a good indicator of cellular redox balance [88]. In *Arabidopsis* and *Oryza sativa*, Cd tolerance is increased through increased expression of genes responsible for GSH synthesis [94]. However, GR activity can be compromised as HM can deplete GSH within plant cells [95], altering the ROS balance [96].

Considering this, it was expected that *GR* and *GST* expression would increase in plants grown in Sb. The results obtained for *GR* expression seem reasonable at 14 days of exposure to the metalloid, once its expression at the root level was higher than that of the control plants (*p* < 0.05, 0.5 mM Sb, *GR_cyt*). The increase in expression at t3 is consistent with the enzymatic activity results described by Espinosa-Vellarino et al. [41], where higher activity was obtained in the roots of plants under Sb stress. Possibly, this increase may be due to the fact that at t3, the plant is trying to generate the maximum GSH possible to evade the stress generated by Sb, especially in the root, which is the organ where this metalloid is preferentially accumulated in. Therefore, these results are also in agreement with the results mentioned in previous sections. That is, *GR* must be continuously expressing and acting to regenerate GSH, trying to defend the plant from the toxic compound [95]. In leaves, there is not a clear expression pattern as in roots, although a reduction in the expression levels using two different Sb concentrations was observed for the *GR_chl* isoform.

In this work, we analyzed the expression pattern of two GST already characterized in Solanaceae. Both of them are involved in detoxification processes and interact with phytohormones such as auxins, cytokinins, salicinates, methyl jasmonate and ethylene [97]. Therefore, their expression is not expected to be lower in control plants considering the number of processes in which they participate. *GST* shows a significant increase in its expression in the leaves of plants grown in Sb at t1 compared to non-exposed plants. At t2, expression levels decrease considerably, showing similar levels to those determined in control plants. In the roots, its expression increases until it reaches the highest value at t3, although no significant differences were found. We can relate these *GST* expression results to those obtained for *DHAR*. GST as DHAR requires GSH for its activity, and in the case of GST, GSH is required to catalyze their binding to Sb. The level of GSH decreases because it is not generated at the same rate as it is expended to alleviate Sb toxicity, even though *GRs* are being expressed and active [41]. Increased *GST* expression is not required as there is no substrate available. In the case of *DHAR*, the level of expression is increased, but not its activity, which might be due to a lack of GSH availability. On the other hand, *GST TCHQDs* at t1 in plants subjected to stress have a similar pattern to the one observed for *GST*. However, the most remarkable difference was found at t3 since there were significant differences in 0.5 mM and 1.0 mM-treated leaves and 1.0 mM-treated roots. The functional characterization of TCHQD is not clear, but it is known that its orthologue in *Sphingobium chlorophenolicum* catalyzes the reductive dehalogenation of tetrachlorohydroquinone (TCHQ) and trichlorohydroquinone to degrade the pesticide pentachlorophenol [52]. That is, it participates in processes of toxic compound elimination, but this does not imply that it has a relevant role in the face of stress by this metalloid [52,81]. Knowledge about the action of GSTs is important in plant breeding because agrobiotechnology on GSTs has allowed the development of transgenic plants with greater tolerance to biotic and abiotic stresses, such as drought or salt stress [81,98,99]. In *Arabidopsis*, *GST* overexpression increases tolerance to Al, Cu, As, Cd and Cr with low levels of peroxidation [100,101].

The expression pattern of *SOD*, *APX*, *GR_chl* and *GST* under different Sb concentrations at the latest time has already been analyzed in a previous study in roots and mature leaves [41]. The expression patterns obtained for roots in the previous study do not show any differences with the present, except for the expression of the *GR* at 0.5 mM, which is higher than in control plants and lower than in plants exposed to 1.0 mM Sb. However, no notable differences were found in the foliar analysis. In all the enzymes mentioned, the patterns obtained in this work differ from those obtained in [41]. In fact, *SOD*, *GR* and *APX* after 14 das have a lower expression pattern in leaves than in the control plants. Only *GST* seems to show a similar pattern, although at 1.0 mM Sb its expression is higher than that of the control plant, contradicting the result obtained in [41]. These results may be due to the change that has been made in relation to the type of leaf that was collected for the study. In [41], mature leaves (third leaf) were analyzed, while in this work young leaf was always tested. Based on the results, we can elucidate that, under the same stress and exposure time, it seems that the foliar response depends on the leaf stage. The mature leaves had to adapt to a change in conditions; therefore, they began to develop in hydroponic cultivation with no Sb, and for this reason it is possible that the expression pattern of *SOD* and *APX* at t1 is similar to that obtained in [41], 14 days under Sb exposure. On the contrary, differences were encountered in the *GR* and *GST* expression levels, perhaps because their action begins later than that of *SOD* or *APX*. Young leaves show a different expression pattern, maybe due to the fact that this type of tissue developed after stress induced by Sb. Another factor to consider is that the third leaf was exposed to Sb stress longer than the young leaf tested. In this work, we aimed to analyze young leaves as we wanted to determine how stress affects development of some organs. Therefore, it is interesting to know that depending on the foliar development stage under stress and the time it has been exposed to this stress, the expression pattern of these enzymes can vary.

### 3.3. Interaction of GSNOR Expression and Root NO Content

GSNOR or class III alcohol dehydrogenase (ADH3) has an important role in RNS metabolism, in the homeostasis of intracellular NO levels and in the control of the transnitrosation balance between S-nitrosylated proteins and GSNO. GSNOR reduces GSNO with NADH as a coenzyme to produce GSSG and ammonium [102,103,104]. GSNOR is located in the nucleus (excluding the nucleolus), cytosol, peroxisomes, chloroplasts, and mitochondria [105,106].

Besides the significant difference observed in leaves treated with 0.5 mM Sb, we could not appreciate any other significant differences between treated and non-treated plants at any time point. Hence, we assumed that the *GSNOR* expression levels do not change due to the Sb stress. In fact, in other studies, authors reported expression in *GSNOR* in plants free of any stress (Airaki et al. [107], in *C. annum*; Martínez et al. [108], in *Arabidopsis;* or Kubienová et al. [109] and Jahnová et al. [110], in tomato). GSNOR can be regulated at both transcriptional and post-translational levels. Regulation of GSNOR contributes to the adjustment of NO signaling in plants. On the one hand, reversible oxidative modification of GSNOR cysteine residues is known to inhibit its enzymatic activity in vitro, suggesting possible forward crosstalk of RNS and ROS signaling at this point [106,111]. NO biosynthesis in plants depends on the site and nature of the stimulus that triggers NO production [112,113]. NO can be produced by oxidative or reductive pathways, enzymatic or non-enzymatic reactions [112,113]. The increase in the NO content and the changes in the expression and activity of *GSNOR* show the participation of RNS in response to Sb. Espinosa-Vellarino et al. [41] mention that Sb altered the balance of ROS. Mainly, these authors describe an increase in H_2_O_2_ concentration that led to an increase in NO to limit the induced redox imbalance. Our results show that SOD expression increases from t1 to t3. Could it be due to the increase in H_2_O_2_ triggered by Sb? A high amount of H_2_O_2_ is capable of producing an increase in NO in the root. It is probable that the excess of H_2_O_2_ is also used for the synthesis of lignin in the cell walls in the roots and as an immobilization system for Sb in the leaves. Through mutants *nox1* (overproduction of NO with high level of l-arginine and l-citrulline) and *gsnor1-3* (reduced activity of GSNOR and high level of NO, nitrate and S-nitrosothiols) in *A. thaliana* exposed to toxic Cu concentrations [102,114], it was possible to determine that a high level of NO due to a reduced activity of GSNOR allows the plant increasing its sensitivity under mild stress conditions and favors tolerance under severe stress conditions [114]. The *gsnor1-3* mutant had a high level of S-nitrosothiols and a higher tolerance to selenium (Se) [115]. However, another study reports an increase in GSNOR activity and NO content in rice plants under toxic concentrations of Al [116]. In Solanaceae (such as potato), an increase in NO without changes in the activity of GSNOR in the root has been described [117], and the results are consistent with ours in tomato. In *Pisum sativum* under Cd toxicity, there is an increase in NO along with a decrease in GSNOR activity [118]. Given this divergence of the *GSNOR* pattern and the NO content, we could conclude that a plant’s response to HM toxicity depends on the plant species and the type of metal/metalloid.

### 3.4. Phylogenetic Analysis and GST and GR Isoform Peptide Sequence and Motif Analysis

We carried out an in-depth bioinformatic study on GR and GST peptide sequences and motifs, as both enzymes play a key role in the defense and detoxification mechanism of plants against abiotic stresses such as HM or metalloids [46,119]. The phylogenetic analysis and the alignment of the peptide sequences of GR and GST allow us evaluation of the evolution of the functional domains of these enzymes in tomato and other members of the Solanaceae family (*C. annuum* and *S. tuberosum*) when comparing them with the orthologs described in reference species of dicots (*Arabidopsis*) and monots (*Oryza sativa* and *Z. mays*). Recently, different studies such as that of Bölükbaşı [42] have shown that GR and GST can be used in plant breeding as possible sources of tolerance.

Looking at our results, we can conclude that there is a high conservation of functional domains between monots and dicots, especially a high conservation of all the residues and motifs described for both GR isoforms and GSTs.

Considering the phylogenetic analysis of Solanaceae GST TCHQD, this isoform seems to diversify early from the rest of the dicots (*Arabidopsis*). Therefore, these differences are found in regions that may determine specific characteristics within the Solanaceae family, although much information is lacking. We hypothesize that the origin of this divergence is due to gene duplication processes. These processes play an important role in the evolution of plants and can serve for the acquisition of new functions [120]. The duplication processes can be of a complete genome, of a single gene or of genome regions [96,121]; however, the single duplication of a gene would be enough for the appearance of new genes [122]. One type of gene duplication is tandem duplication involving two or more homologous genes adjacent to each other in the genome [123]. For both Brassicaceae (*Arabidopsis*) and monots, tandem duplications have occurred. In fact, these types of regions represent a significant proportion of their genomes: 17% in *Arabidopsis* [124], 14% in *Oryza sativa* [125] or 35% in *Z. mays* [126]. Solanaceae is not an exceptional case. Within this family, there are many duplicated chromosome fragments. In addition to the duplication process experienced by all dicots, about 65 million years ago, the Solanaceae species had another duplication event [127]. Duplicated genes were subsequently separated. Therefore, this would explain why the GR and GST regions are conserved in some residues. The conserved residues might allow them to perform their function, but divergence in those residues in other families or species might lead to different behaviors when exposed to the same stresses. This could explain the existence of the different types of GR or GST described.

## 4. Materials and Methods

Tomato seeds (*S. lycopersicum* L. cv. “Tres Cantos”, Mascarell^®^, Benissoda, Spain) were used. For germination, the seeds were placed on wet filter paper in the dark at 28 °C for 48–72 h. When germinated, the seedlings were transferred to a plastic container (30 cm × 20 cm × 10 cm; width, length and height, respectively) filled with sterile vermiculite and grown at 24–26 °C, 85% relative humidity and 350 μmol m^−2^ s^−1^ under a long-day photoperiod (16 h/8 h day/night). After 5 days, the plants were transferred to hydroponic culture for 14 days in a lightweight polypropylene container (30 cm × 20 cm × 10 cm; 4 plants per-container). Hydroponic cultivation allowed no retention of Sb by different elements that occur in standard soil. The plants were continuously aerated, and the same conditions were used (with the exception for relative humidity, 50%). The plants were treated with a basal nutrient solution consisting of 4 mM KNO_3_, 3 mM Ca (NO_3_)_2_ 4H_2_O, 2 mM MgSO_4_ 7H_2_O, 6 mM KH_2_PO_4_, 1 mM NaH_2_PO_4_ 2H_2_O, 10 mM ZnSO_4_ 7H_2_O, 2 mM MnCl_2_ 4H_2_O, 0.25 mM CuSO_4_ 5H_2_O, 0.1 mM Na_2_MoO_4_ 2H_2_O, 10 mM H_3_BO_3_, and 20 mM NaFeIII-EDTA (adjusted pH at 5.8). For Sb treatment, the basal solution was supplemented with KSb (OH)_6_ to final concentrations of 0.0 (control), 0.5 mM, and 1.0 mM Sb (pH 5.8). The hydroponic culture was refreshed with the corresponding treatments every 4 days. A total of 15 plants were analyzed for each treatment with three biological replicates.

Different antimony concentrations were applied (0.25–1.25 mM) to *Dittrichia*, a plant with great potential for phytoremediation, as described in previous studies [40]. Concentrations of 0.5 and 1.0 mM were selected because a concentration of 0.5 generated intermediate phenotypic damage and that of 1.0 a more severe one; however, 0.25 mM doses were not applied as they lead to plant death. The form of Sb applied was antimoniate (V) as it is the least toxic.

### 4.1. Determination of Leaf Areas

Photographs of each plant and all their leaves were taken by the DXM 1200C camera (Nikon^®^ Corp., Tokio, Japan). Each photograph was analyzed with the free software Fiji Image J 2.3.2 (https://imagej.net/Fiji (accessed on 1 June 2023)). At least 15 different plants were counted and repeated three times. The area was calculated using the polygon and measure tools and all the leaves of each plant were considered for the total area (photographs were recorded in 28-day-old plants).

### 4.2. Determination of Sb Contents

To determine Sb concentrations in roots and shoots, samples were maintained at 70 °C for 72 h, then crushed in a ceramic mortar, and, after acid digestion, Sb was assayed by inductively coupled plasma mass spectrometry (ICP-MS) (114). The samples were subject to an acid oxidizing digestion, with HNO_3_ (65%) and H_2_O_2_ (30%), in a high-pressure Start D microwave oven from Milestone. A digestion program was used in a range of powers and temperatures, with a total duration of 1 h 50 min (5 min at 500 W and 100 °C,15 min at 700 W and 170 °C, 5 min at 900 W and 200 °C, 15 min at 900 W and 200 °C, and finally cooling down during 1 h 10 min). Antimony content was assayed in an UNE-EN ISO/IEC 17025 certified laboratory using an Agilent ICP-MS instrument (Agilent Technologies^®^, Santa Clara, CA, USA). Quality control of Sb assays was performed by calibrations against certified commercial Sb standards, and drift controls were carried out to ensure reliability of analyses.

### 4.3. RNA Extraction and cDNA Transformation

Roots and young leaves from plants grown under the different experimental conditions (control, 0.5 mM Sb, and 1.0 mM Sb) were collected. The plant material was frozen in liquid nitrogen and the RNA was extracted and purified using the Spectrum Plant Total RNA kit (Sigma-Aldrich^®^, Milwaukee, WI, USA) and RNase-Free DNase (Qiagen^®^, Hilden, Germany) Nº 79254). The quantity and quality of RNA in the resulting solution were determined using an Eppendorf Biophotometer D30 (Eppendorf^®^, Hamburg, Germany). The integrity of the extracted RNA was assessed by gel electrophoresis with 1× TBE buffer Invitrogen^TM^, Waltham, MA, USA) and with GelStar^TM^ (LonzaRockland Inc., Rockland, ME, USA) as intercalating agent, loading 2.5 µL of RNA from each sample with 10 µL of RNase-free water (Invitrogen^TM^, Waltham, MA, USA) and 2 µL of a loading buffer (Invitrogen^TM^, Waltham, MA, USA). The results were visualized in an Azure Imaging System C200 transilluminator (Azure Biosystems^®^ Inc., Dublin, CA, USA). Samples of 1–2 µg of purified RNA were reverse-transcribed with the High-Capacity cDNA Reverse Transcription kit from Applied Biosystems^®^ (Foster City, CA, USA) and the reaction was carried out in a thermocycler (Eppendorf, Hauppauge, NY, USA) with a first stage at 25 °C for 10 min, followed by a stage at 37 °C for 120 min, and a final stage at 85 °C for 5 min, obtaining single-stranded cDNA.

### 4.4. PCR and qRT-PCR

Previously, a PCR was performed using the Taq Hot Start Premix version of Takara Bio Inc. (Shiga, Japan) and the primers shown in Appendix A to verify that each pair of primers did not produce non-specific amplifications. The PCR protocol was as follows: activation of polymerase at 95 °C for 5 min; thermal cycling with 3 steps: 95 °C for 30 s; 57 °C for 30 s, elongation 72 °C for 30 s for 40 cycles; and a final extension at 72 °C for 10 min. The PCR products were revealed in a 1% agarose by gel electrophoresis with a 1× TBE buffer Invitrogen^TM^, Waltham, MA, USA and with GelStar^TM^ (LonzaRockland Inc., Rockland, ME, USA) as an intercalating agent. The results were visualized in an Azure Imaging System C200 transilluminator (Azure Biosystems^®^ Inc., Dublin, CA, USA).

Real-time amplification was monitored with SYBR Green (Thermo Fisher Scientific, Waltham, MA, USA) on a QuantStudio 1 amplification and detection instrument (Applied Biosystems, Thermo Fisher Scientific R, Waltham, MA, USA). Each target gene was paired with two different reference genes: *Actin* [Solyc04g011500.2.1] and *β-tubulin* [Solyc04g081490.2.1]). Both were used as housekeeping genes because in studies related to abiotic stress [128], they are recommended to normalize gene expression of target genes. The expression of each target gene relative to the expression of the reference gene was calculated using the 2-ΔΔCt method.

### 4.5. Statistical Analyses

For nitric oxide content values in the root and the values obtained leaf area, Student’s *t*-test was performed for statistical analysis with Microsoft Excel (Microsoft^®^ Corp., Alburquerque, NM, USA). Significant differences are presented as * *p* < 0.05 and ** *p* < 0.01. The data presented are the means, and error bars denote SD of at least 15 replicates obtained from three independent experiments.

For all the values obtained in molecular studies, a Shapiro–Wilk test was performed to determine whether the data had a normal distribution. A one-way ANOVA test or a Kruskal–Wallis test were performed with RStudio version-4.3.2 (RStudio, Inc., Boston, MA, USA). In both cases, these tests were used for statistical significance (significant differences are presented as * *p* < 0.05 and ** *p* < 0.01). The data presented are the means, and error bars denote SD of at least 15 replicates obtained from three independent experiments.

### 4.6. Visualization and Determination of Nitric Oxide (NO)

Control and Sb-treated plant roots (20 mm) were incubated for 60 min in the dark at 25 °C with 10 μM DAF-2DA in 10 mM Tris HCl, pH 7.4, and washed following the method described by Valderrama et al. [129]. The whole roots were placed on a slide and examined under a fluorescence microscope (Zeiss Axioplan, Oberkochen, Germany). At least five roots were tested under each experimental condition and five independent repetitions were analyzed. Images were processed and analyzed using the ImageJ program, and fluorescence intensity was expressed in arbitrary units.

### 4.7. Bioinformatic Analysis

The coding and peptide sequences of the enzymes GR and GST were analyzed (Appendix A). The sequences were obtained in the GRAMENE (www.gramene.org (accessed on 1 June 2023) and Solgenomics (https://solgenomics.net/ (accessed on 1 June 2023)) [130] databases. The analysis consisted of a global alignment of the nucleotide coding sequence and the amino acid sequence using Geneious Prime 2023.1.1 software (Geneious Biologics 2023 (https://www.geneious.com/biopharma (accessed on 1 June 2023))). For these alignments, the Blosum90 matrix (BLOcks of amino acid Substitution Matrix with a maximum identity of 90%) was used with “free end gaps”, and the genetic distance was calculated with the Jukes–Cantor model [131]. Protein domains, motifs and conserved residues were identified in the current literature and mapped to the deduced proteins. Phylogenetic analysis was performed using the Jukes–Cantor model and Neighbor-Joining as a statistical model [132]. The interaction of the target proteins was predicted using the STRING version 11.0 database (https://string-db.org/ (accessed on 1 June 2023)) [133].

## 5. Conclusions

Overall, we can conclude that the enzymes that participate in the AsA-GSH cycle present changes in their coding gene expression levels under the stress caused by the presence of the metalloid Sb. In addition, when the results of leaf expression at t3 were compared with previous studies [41], we observed changes in gene expression in mature and young leaves under the same stress condition. These differences in the expression pattern of the target genes analyzed can help understand how plants modify their behavior to adapt to new development and growth conditions.

In our study, we observed that GR isoforms present similar expression patterns and high conservation of their functional motifs with other monocots and dicots. However, we appreciated the differences between the GST analyzed. Therefore, the GST TCHQD isoform coding gene always presents a lower expression in the organs analyzed compared to the non-treated plants and to other GST, especially for the 1.0 mM Sb treatment. Moreover, this isoform lacks some conserved residues. Therefore, further studies will shed light on their capacity to cope with certain stresses even lacking those conserved regions. Finally, no relation was found between *GSNOR* expression and NO content in tomato roots under Sb stress.

## Figures and Tables

**Figure 1 plants-13-00609-f001:**
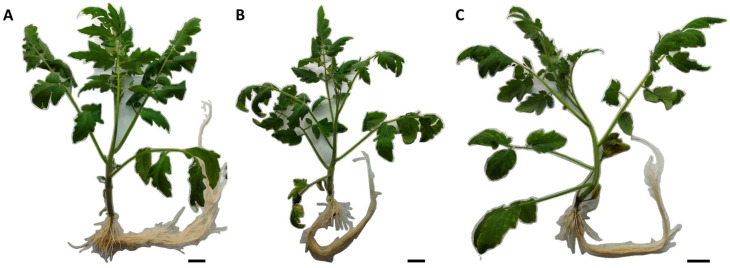
Phenotypic appearance at final time (t3, 14 das) of control plants (**A**), tomato plants with 0.5 mM Sb treatments (**B**) and tomato plants exposed to 1.0 mM of the metalloid (**C**). Scale bar—3 cm.

**Figure 2 plants-13-00609-f002:**
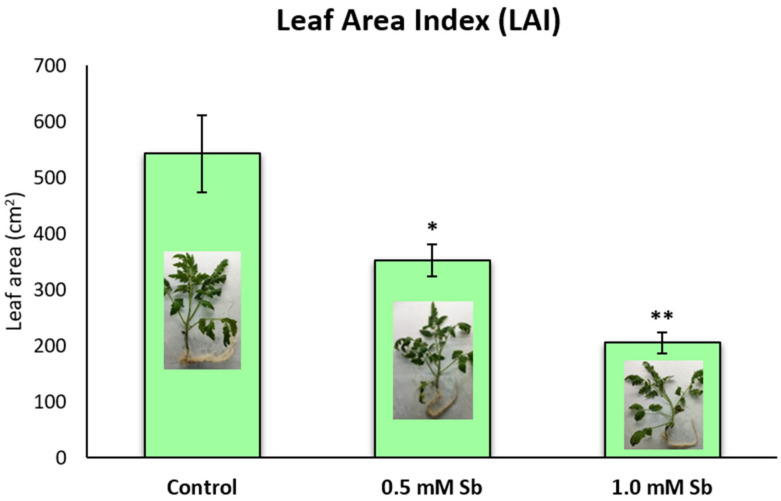
Leaf area at final time (t3, 14 das) of control tomato plants and subjected to treatments of 0.5 mM and 1.0 mM of Sb. At least 15 plants per treatment were analyzed. Values are means and error bars denote ± standard deviation (SD). Differences from control values were significant at * *p* < 0.05, ** *p* < 0.01 (Student’s *t*-test).

**Figure 3 plants-13-00609-f003:**
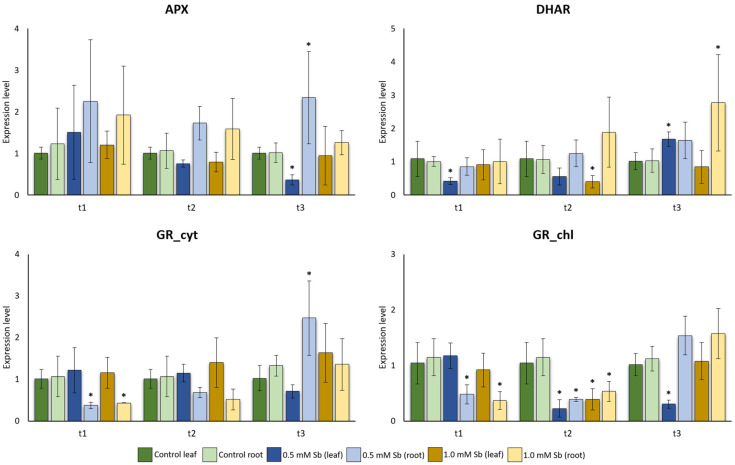
Effect of Sb on the levels of *APX*, *DHAR*, *GR_cyt* and *GR_chl* gene expression in roots and leaves of tomato plants. Values are means and error bars denote ± standard deviation (SD). Differences from control values were significant at * *p* < 0.05, ** *p* < 0.01 (Kruskal–Wallis *t*-test).

**Figure 4 plants-13-00609-f004:**
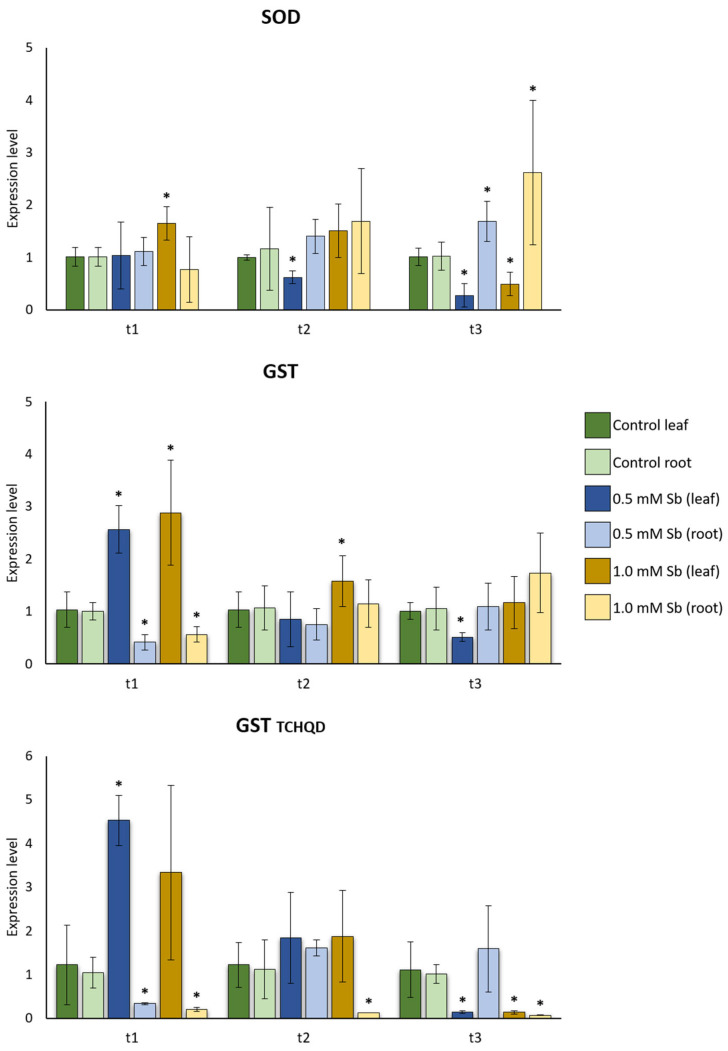
Effect of Sb on the levels of *SOD*, *GST* and *GST TCHQD* gene expression in roots and leaves of tomato plants. Values are means and error bars denote ± standard deviation (SD). Differences from control values were significant at * *p* < 0.05, ** *p* < 0.01 (Krukal–Wallis *t*-test).

**Figure 5 plants-13-00609-f005:**
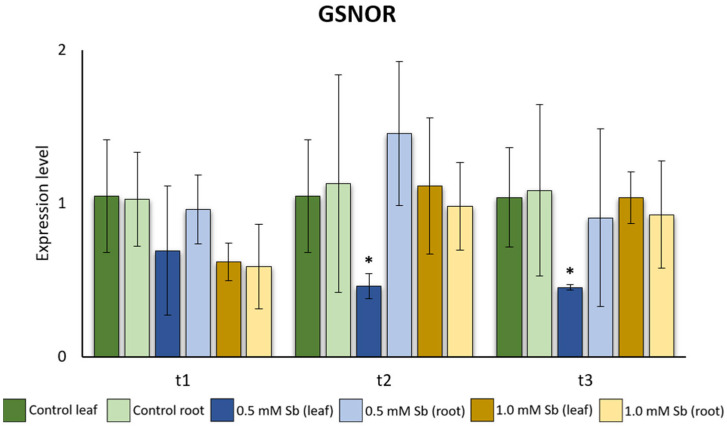
Effect of Sb on the levels of *GSNOR* gene expression in roots and leaves of tomato plants. Values are means and error bars denote ± standard deviation (SD). Differences from control values were significant at * *p* < 0.05, ** *p* < 0.01 (Kruskal–Wallis *t*-test).

**Figure 6 plants-13-00609-f006:**
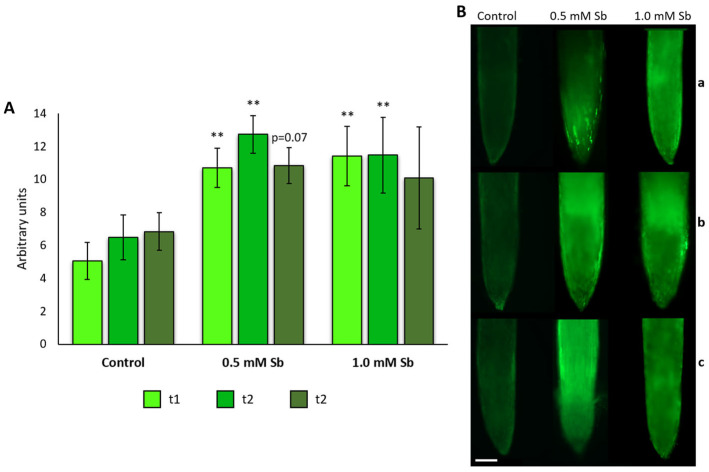
NO content was detected by fluorescence. NO content detected in the roots of control plants and in plants exposed to 0.5 mM Sb and 1.0 mM Sb (**A**). Fluorescence intensity levels were quantified in arbitrary units. Values are means and error bars denote ± standard deviation (SD). Differences from control values were significant at * *p* < 0.05, ** *p* < 0.01 (Student’s *t*-test). Micrographs of longitudinal sections of roots of control plants, plants exposed to 0.5 mM of Sb and plants exposed to 1 mM of Sb (**B**) after 1 das (t1, **a**), after 7 das (t2, **b**) and after 14 das (t3, **c**). Scale bar—200 µm. At least 15 roots were tested for each experimental condition and 3 independent repetitions were analyzed.

**Figure 7 plants-13-00609-f007:**
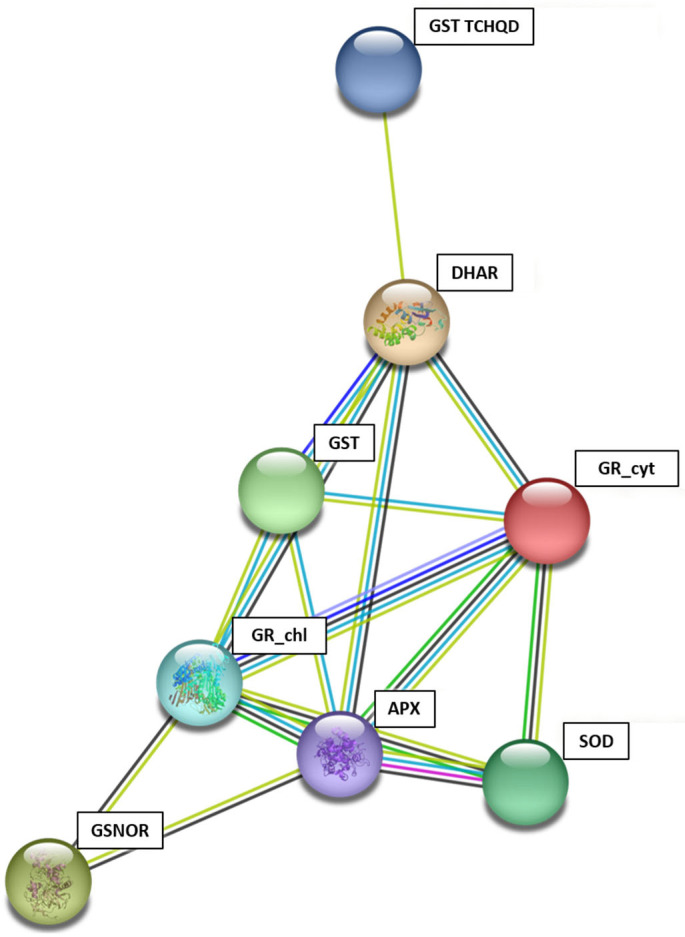
Protein–protein interaction network among APX, DHAR, GR_chl, GR_cyt, GSNOR, GST and GST TCHQD enzymes developed using STRING: Red—fusion evidence. Green—neighborhood evidence. Blue—co-occurrence evidence. Purple line—experimental evidence. Yellow line—text-mining evidence. Light blue line—database evidence. Black—co-expression evidence.

**Figure 8 plants-13-00609-f008:**
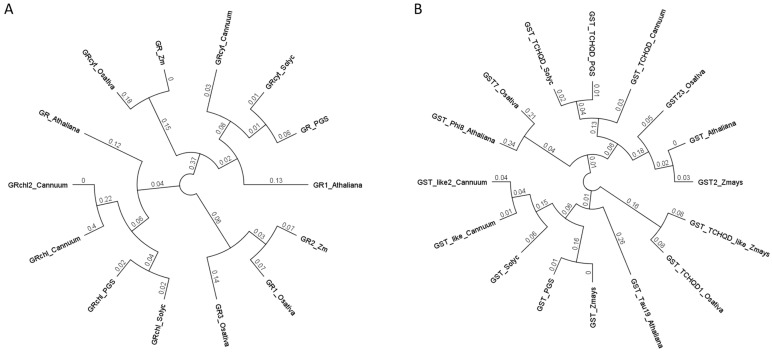
Sequence-based phylogenetic relationships amoung GR_chl (**A**), GR_cyt (**A**), GST (**B**) and GST_TCHQD (**B**) in *Arabidopsis*, *Zea may* L.*s*, *Oryza sativa* L. spp. *japonica* and Solanaceous species. Deduced protein sequences were obtained from Gramene and used for a global alignment with the Geneious program using Cost Matrix Blosum90 in the GENEIOUS platform. The phylogenetic tree was constructed with the Geneious tree builder. Athaliana, *Arabidopsis thaliana* L.; Zm, *Z. mays*; Osativa, *Oryza sativa* L. spp. *japonica*; Cannum, *Capsicum annuum* L.; PGS, *Solanum tuberosum* L.; Solyc, *Solanum lycopersicum* L.

**Figure 9 plants-13-00609-f009:**
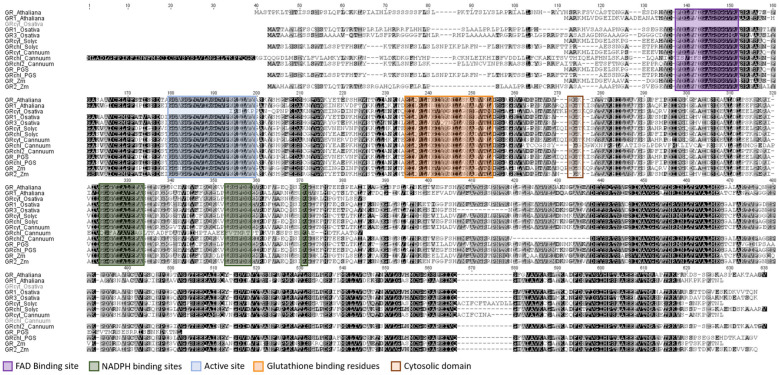
GR peptide sequence alignment: isoforms from different species of the Solanaceae family and monocot and dicot model species. Deduced protein sequences were obtained from Gramene and used for a global alignment with the Geneious program using Cost Matrix Blosum90 in the GENEIOUS platform. The FAD binding domains (purple), NADPH binding sites (green), glutathione binding residues (orange) and cytosolic domain (brown) are represented.

**Figure 10 plants-13-00609-f010:**
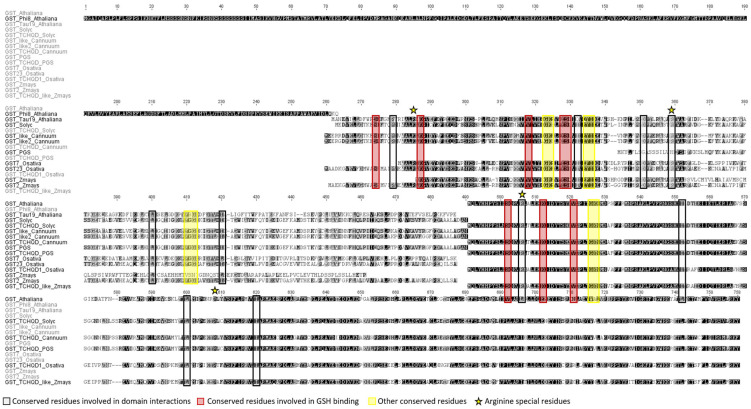
GST peptide sequence alignment: isoforms from different species of the Solanaceae family and monocot and dicot model species. Deduced protein sequences were obtained from Gramene and used for a global alignment with the Geneious program using Cost Matrix Blosum90 in the GENEIOUS platform. The conserved residues involved in domain interactions (black), the conserved residues involved in GSH binding (red) and other conserves residues (yellow).

## Data Availability

Data are contained within the article and Appendix A.

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
