# Peer review of "Enzymes Involved in Antioxidant and Detoxification Processes Present Changes in the Expression Levels of Their Coding Genes under the Stress Caused by the Presence of Antimony in Tomato"

_plants, 2024, doi:10.3390/plants13050609_

Round 1

Reviewer 1 Report

Comments and Suggestions for Authors

This study by Francisco et al., is a continuation of work on their previous publication from the same group as the following.

Espinosa-Vellarino, F. L.; Garrido, I.; Ortega, A.; Casimiro, I.; Espinosa, F. Effects of antimony on reactive oxygen and nitrogen species (ROS and RNS) and antioxidant mechanisms in tomato plants. Front. Plant Sci. 2020,

Even they added the data at two more different time points for the test as compared to the previous publication, and they added a few more bioinformatic data after that; It makes the novelty weak. Some data and logic lines (Figure1 to Figure 6) are highly similar to the above previous publication.

1)     The results here in Figure 1 are quite similar to the above publication in 2020, it looks like the same treatment, but just different individual plants.

2)     In Figure 3 in this manuscript, they tested the expression level instead of genes encoded enzyme’s activity in Figure 2 in the publication in 2020.

3)     In Figure 4 in this manuscript, they tested the transcript level of SOD, APX, GST, GST TCHQD instead of SOD, APX, GR, and GST gene expression in the publication in 2020.

4)     in the above publication in 2020, they tested the content of nitric oxide too in figure 4, in this manuscript, the author just added more time points in Figure 6.

Other comments:

1) The leaf area data at t3 time point in Figure 2 looks didn’t match the phenotype in Figure 1C.

2) Too much citation used in this manuscript.

In addition, there are a few small formatting errors, such as the species name should be italic, “Arabidopsis thaliana (dicot) and Oryza sativa” in the text from 291 to 307. It needed to be checked carefully throughout the manuscript.

Comments on the Quality of English Language

The manuscript is okay to read. 

Author Response

Thank you very much for your advice and mentions to improve our manuscript. In this manuscript we wanted to complement previous work from a molecular point of view and determine the performance of GR and GST isoforms in the face of this stress.

1)     The results here in Figure 1 are quite similar to the above publication in 2020, it looks like the same treatment, but just different individual plants.

2)     In Figure 3 in this manuscript, they tested the expression level instead of genes encoded enzyme’s activity in Figure 2 in the publication in 2020.

3)     In Figure 4 in this manuscript, they tested the transcript level of SOD, APX, GST, GST TCHQD instead of SOD, APX, GR, and GST gene expression in the publication in 2020.

4)     in the above publication in 2020, they tested the content of nitric oxide too in figure 4, in this manuscript, the author just added more time points in Figure 6.

Other comments:

1) The leaf area data at t3 time point in Figure 2 looks didn’t match the phenotype in Figure 1C.

2) Too much citation used in this manuscript.

In addition, there are a few small formatting errors, such as the species name should be italic, “Arabidopsis thaliana (dicot) and Oryza sativa” in the text from 291 to 307. It needed to be checked carefully throughout the manuscript.

  1. The tomato plants were grown in hydroponic cultivation at different concentrations of antimony but the final time was shorter (14 days). We wanted to show that there were no phenotypic changes as a consequence of factors other than antimony.

2) We wanted to test whether these differences at the level of enzymatic activity corresponded to changes in the expression of those enzymes. Our intention is to establish a correlation between both measurements and thus make a dissertation if there are differences between gene activity and expression and how it evolves from short times (1 day) to long times subjected to stress (14 days).

3) Our goal in this manuscript is to delve into the influence of two isoforms of GST. Thus, determining if any isoform is more active, at the level of expression, in the tomato plant's fight against the stress promoted by antimony.

4) Exactly we wanted to test how NO production evolved at different times and establish a connection with GSNOR activity at the foliar and root level.

It is possible that the selected image does not seem very clarifying to you, but we guarantee that the data and evaluations on the plants analyzed in various experiments are reliable and if you require them we can provide them.

We are aware that we use many refrences, but we have been working on this type of essay for a long time and we wanted to demonstrate that we have carried out an exhaustive search.

We deeply regret these errors. Thanks for your recomendation.

Reviewer 2 Report

Comments and Suggestions for Authors

The manuscript presented by Espinosa-Vellarino et al. reports of studies performed in tomato under Sb stress. As a first observation, the manuscript still contains "instructions" from the manuscript template and this is quite confusing in some places.

The paper contains too many references, 175 for such a paper is not unacceptable. Also, some of them do not seem completely appropriate: for example 1, 2, 8, 12, 13, 40. 41, 57, 58.

Experimental design: The reason for choosing concentrations and timing of Sb stress is not given. Also there is no evaluation of Sb content in plant organs, no measurement of biomass or oxidative  stress status.

Figure legends are not sufficiently informative. Many details essential for understanding the experiments are missing.

The authors pay no attention to use of taxonomic indications for plants, they do not use common conventions, they do not report botanical authorities consistently.

The discussion is too long. It is also highly speculative based on the data available and the lack of analysis for Sb content in plant organs.

The description of methods lacks many details essential for reproducibility.

L130 - the authors use too many times the expression "on the other hand"

L135 and others: normally the first level of significance is p<0.05 - is this a mistake or is it really p<0.005? Same for 0.001, is it 0.01? SEE L756

L139 and following: a short explanation on how the gene expression changes were evaluated is necessary, since the description of methods comes later.

L167 Figure 3, Figure 4, Figure 5 are difficult to read, the bars in the graph do not allow an immediate comparison between treatments for the same organs. It is also not clear how the statistics was performed, separately for each organ or treatment.

L241 In figure 6 the timing of experiments is indicated differently than in other figures, it is confusing.

L260 Figure 7 cannot be understood, the legend is insufficient to understand the meaning of colours and symbols

L263 and following: this part of the paper is not coherent with the experiments and the purpose of the study. L311-316, L617 and following, do not justify why this analysis has been included.

L321 and following: for no apparent reason all figures are repeated here

L448 according to conventions, figures should not be cited in the discussion

L717 I think that now M must be substituted with mol/L, as a standard measure unit

L726 provide details

L744-749 too little details about computations of gene expression levels

L796 I do not understand "Patents"

L797-8 there are problems in the supplementary tables.

L799, L801 Details given are insufficient

Comments on the Quality of English Language

The English level is not bad but at times there are mistakes which should be corrected.

Author Response

Thank you very much for your advice and mentions to improve our manuscript. In this manuscript we wanted to complement previous work from a molecular point of view and determine the performance of GR and GST isoforms in the face of this stress. Below we try to answer your questions and correct the errors you report to us.

The paper contains too many references, 175 for such a paper is not unacceptable. Also, some of them do not seem completely appropriate: for example 1, 2, 8, 12, 13, 40. 41, 57, 58.

  • We are aware that we use many refrences, but we have been working on this type of essay for a long time and we wanted to demonstrate that we have carried out an exhaustive search.

Experimental design: The reason for choosing concentrations and timing of Sb stress is not given. Also there is no evaluation of Sb content in plant organs, no measurement of biomass or oxidative  stress status.

  • This work seeks to deepen the knowledge obtained in previous work for the group, which is why we have information on concentrations, times of exposure to stress, etc.

Figure legends are not sufficiently informative. Many details essential for understanding the experiments are missing.

  • We would like to know what information you think is most convenient to add to the legends to improve them.

The authors pay no attention to use of taxonomic indications for plants, they do not use common conventions, they do not report botanical authorities consistently.

  • We have made the appropriate changes to correct these errors. We regret these errors.

The discussion is too long. It is also highly speculative based on the data available and the lack of analysis for Sb content in plant organs.

  • The content of Sb, how it is stored in the plant, has already been analyzed by our group in previous works, which is why we did not want to include them again because they did not contribute anything new. If you need this data or think it is convenient to put it in an annex, we will do so.

The description of methods lacks many details essential for reproducibility.

- We don't know what part of the materials and methods you want us to delve deeper into. Please give us a clarification to correct it.

L130 - the authors use too many times the expression "on the other hand"

  • We have made editorial changes to correct this.

L135 and others: normally the first level of significance is p<0.05 - is this a mistake or is it really p<0.005? Same for 0.001, is it 0.01? SEE L756

  • We have made editorial changes to correct this.

L139 and following: a short explanation on how the gene expression changes were evaluated is necessary, since the description of methods comes later.

  • We have written the expression levels based on other manuscripts where they analyze the expression levels in a similar way to us.

L167 Figure 3, Figure 4, Figure 5 are difficult to read, the bars in the graph do not allow an immediate comparison between treatments for the same organs. It is also not clear how the statistics was performed, separately for each organ or treatment.

  • We did a statistical treatment to determine if there were significant differences between different treatments in each plant organ and at each time separately.

L241 In figure 6 the timing of experiments is indicated differently than in other figures, it is confusing.

  • We have made editorial changes to correct this.

L260 Figure 7 cannot be understood, the legend is insufficient to understand the meaning of colours and symbols

  • We have made editorial changes to correct this.

L263 and following: this part of the paper is not coherent with the experiments and the purpose of the study. L311-316, L617 and following, do not justify why this analysis has been included.

  • The phylogenetic analysis and the protein motifs of the GR and GST isoforms help us to know if there are variations between Solanaceae and other known species of interest. This assesses whether there are intraspecific or interspecific variations and which provides these plants with better tools to face this type of stress.

L321 and following: for no apparent reason all figures are repeated here

  • We thought that we had to include all the figures in that section. Thanks for your advice.

L448 according to conventions, figures should not be cited in the discusión

  • Thank you once again for your advice, we have made the corresponding changes.

L717 I think that now M must be substituted with mol/L, as a standard measure unit

  • In other manuscripts of this editorial we were suggested to use M.

L726 provide details L744-749 too little details about computations of gene expression levels. L799, L801 Details given are insufficient

  • We do not know how to satisfy those details you refer to. If you can let us know more information we can correct it.

L796 I do not understand "Patents"

  • We have made editorial changes to correct this.

Round 2

Reviewer 1 Report

Comments and Suggestions for Authors

Thanks for the author’s explanation, but most of my comments haven’t been addressed. The author just did a little bit of words and formatting revision based on the first version.

Because this study is a continuation of work on their previous publication. Moreover, some data and logic lines (Figure1 to Figure 6) are highly similar to their previous publication.

If the author wants to keep these data, the author at least needs to introduce the previous work before the “Result” section. And the author needs to compare the data which were collected at the different time point here to the previous data, to show/discuss whether they found some new things in the result section or discussion section.

As to “1) The leaf area data at t3 time point in Figure 2 looks didn’t match the phenotype in Figure 1C.” if the bar in Figure 1C is correct, the leaf area of tomato plants exposed to 1 mM Sb is larger than/similar to control plants and the plants under 0.5mM Sb based on the phenotype. If this is due to the individual difference, the error bar in Figure 2 should be larger.  

2) Too many citations used in this manuscript. Remove not directly related references.

There are still a few small formatting errors. Such as Figure citations, some as (figure x), some as (Figure x)

Comments on the Quality of English Language

okay to read

Author Response

Thank you very much for your comments and advice. We have attempted to respond to your requests, we hope that you consider this version of the manuscript to be of sufficient quality for publication. We have referenced previous work we did on tomatoes. Our goal has been to evaluate how the response of the tomato evolves at different times to the stress caused by Sb, since previous studies were only based on analyzing it at 14-21 days, when the plant is already more damaged.

Reviewer 2 Report

Comments and Suggestions for Authors

The authors have changed few numbers and few words. They have not replied satisfactorily to my comments and no changes have been made to the text. The reference to previous data and experiments is not clear at all, and apparently this has not been improved.

The changes to plant species names have not been done professionally and they are still wrong.

As is conventional in scientific papers - methods should be described in such details to allow the reproducibility of experiments by other scientists - here essential details are missing.

The figure legends and table headings should allow the understanding of the content without reading the whole paper, and they must explain everything in the figure, colours, symbols etc. - here it is not like this.

The comments about improving the description of methods and legends have not been considered.

This version is almost exactly like the first version.

Comments on the Quality of English Language

OK

Author Response

Thank you very much for your comments and advice. We have attempted to respond to your requests, we hope that you consider this version of the manuscript to be of sufficient quality for publication.